# Hierarchical Object Representation for Open-Ended Object Category Learning and Recognition

**S.Hamidreza Kasaei, Ana Maria Tomé, Luís Seabra Lopes**
IEETA - Instituto de Engenharia Electrónica e Telemática de Aveiro
University of Aveiro, Averio, 3810-193, Portugal
{seyed.hamidreza, ana, lsl}@ua.pt

## Abstract

Most robots lack the ability to learn new objects from past experiences. To migrate a robot to a new environment one must often completely re-generate the knowledge-base that it is running with. Since in open-ended domains the set of categories to be learned is not predefined, it is not feasible to assume that one can pre-program all object categories required by robots. Therefore, autonomous robots must have the ability to continuously execute learning and recognition in a concurrent and interleaved fashion. This paper proposes an open-ended 3D object recognition system which concurrently learns both the object categories and the statistical features for encoding objects. In particular, we propose an extension of Latent Dirichlet Allocation to learn structural semantic features (i.e. topics) from low-level feature co-occurrences for each category independently. Moreover, topics in each category are discovered in an unsupervised fashion and are updated incrementally using new object views. The approach contains similarities with the organization of the visual cortex and builds a hierarchy of increasingly sophisticated representations. Results show the fulfilling performance of this approach on different types of objects. Moreover, this system demonstrates the capability of learning from few training examples and competes with state-of-the-art systems.

## 1   Introduction

Open-ended learning theory in cognitive psychology has been a topic of considerable interest for many researchers. The general principle is that humans learn to recognize object categories ceaselessly over time. This ability allows them to adapt to new environments, by enhancing their knowledge from the accumulation of experiences and the conceptualization of new object categories [1]. In humans there is evidence of hierarchical models for object recognition in cortex [2]. Moreover, in humans object recognition skills and the underlying capabilities are developed concurrently [2]. In hierarchical recognition theories, the human sequentially processes information about the target object leading to the recognition result. This begins with lower level cortical processors such as the elementary visual cortex and go "up" to the inferotemporal cortex (IT) where recognition occurs. Taking this as inspiration, an autonomous robot will process visual information continuously, and perform learning and recognition concurrently. In other words, apart from learning from a batch of labelled training data, the robot should continuously update and learn new object categories while working in the environment in an open-ended manner. In this paper, "open-ended" implies that the set of object categories to be learned is not known in advance. The training instances are extracted from *on-line* experiences of a robot, and thus become gradually available over time, rather than completely available at the beginning of the learning process.

Classical object recognition systems are often designed for static environments i.e. training (offline) and testing (online) are two separated phases. If limited training data is used, this might lead to non-discriminative object representations and, as a consequence, to poor object recog-

nition performance. Therefore, building a discriminative object representation is a challenging step to improve object recognition performance. Moreover, time and memory efficiency is also important. Comparing 3D directly based their local features is computationally expensive. Topic modelling is suitable for open-ended learning because, not only it provides short object descriptions (i.e. optimizing memory), but also enables efficient processing of large collections.

This paper proposes a 3D object recognition system capable of learning both object categories as well as the topics used to encode them *concurrently* and in an *open-ended* manner. We propose an extension of Latent Dirichlet Allocation to learn incrementally topics for each category independently. Moreover, topics in each category are discovered in an unsupervised fashion and updated incrementally using new object views. As depicted in Fig.1, the approach is designed to be used by a service robot working in a domestic environment. Fig.1(left), shows a PR2 robot looking at some objects on the table. Fig.1(right) shows the point cloud of the scene obtained through

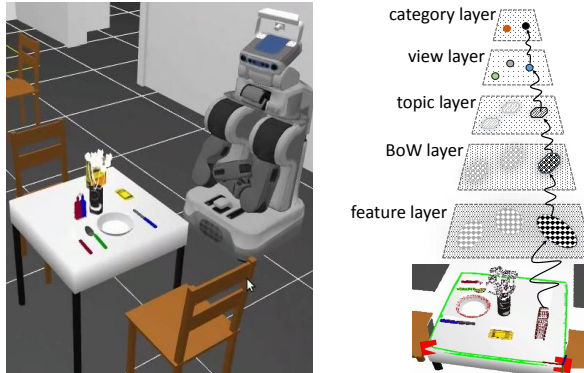

Figure 1: The proposed multi layer object representation being tested on a service robot. It consists of five layers of hierarchy including feature layer, BoW layer, topic layer, object view layer and category layer.

the robot's Kinect and the used representations. Tabletops objects are tracked (signed by different colors) and processed through a hierarchy of five layers. For instance, to describe an object view, in the feature layer, a spin-image shape descriptor [3] is used to represent the local shapes of the object in different key points; afterwards, in the Bag-of-Words (BoW) layer, the given object view is described by histograms of local shape features, as defined in Bag-of-Words models; in the topic layer, each topic is defined as a discrete distribution over visual words and each object view is described as a random mixture over latent topics of the category and stores them into the memory (view layer). Finally, the category model is updated by adding the obtained representation (category layer).

The remainder of this paper is organized as follows. In section2, we discuss related works. Section3 provides a system overview. The methodology for constructing visual words dictionary is presented in section4. Section5 describes the proposed object representation. Object category learning and recognition are then explained in section6. Evaluation of the proposed system is presented in section7. Finally, conclusions are presented and future research is discussed.

## 2 Related work

One of the important tasks in the field of assistive and service robots is to achieve human-like object category learning and recognition. Riesenhuber and Poggio [2] proposed a hierarchical approach for object recognition consistent with physiological data, in which objects are modelled in a hierarchy of increasingly sophisticated representations.

Sivic et al. [4] proposed an approach to discover objects in images using Probabilistic Latent Semantic Indexing (pLSI) modelling [5]. Blei et al. [6] argued that the pLSI is incomplete in that it provides no probabilistic model at the level of documents. They extended the pLSI model calling the approach Latent Dirichlet Allocation (LDA). Similar to pLSI and LDA, we discover topics in an unsupervised fashion. Unlike our approach in this paper, pLSI and LDA do not incorporate class information.

Several works have been presented to incorporate a class label in the generative model [7][8][9]. Blei et al. [7] extend LDA and proposed Supervised LDA (sLDA). The sLDA was first used for supervised text prediction. Later, Wang et al. [8] extended sLDA to classification problems. Another popular extension of LDA is the classLDA (cLDA) [9]. Similar to our approach, the only supervision used by sLDA and cLDA is the category label of each training object. However, there are two main differences. First, the learned topics in sLDA and cLDA are shared among all categories, while we propose to learn specific topics per category. Second, the sLDA and cLDA approaches follow a standard train-and-test procedure (i.e. set of classes, train and test data are known or available in

advance), our approach can incrementally update topics using new observations and the set of classes is continuously growing. There are some topic-supervised approaches e.g. Labeled LDA [10] and semiLDA [11] that consider class labels for topics. On one hand, these approaches need tens of hours of manual annotation. On the other hand, a human can not provide a specific category label for a 3D local shape description (e.g. a spin-images [3]).

There are some LDA approaches that support incremental learning of object categories. The difference between incremental and open-ended learning is that the set of classes is predefined in incremental learning, while in open-ended learning the set of classes is continuously growing. Banerjee et al. proposed [12] online LDA (o-LDA) that is a simple modification of batch collapsed Gibbs sampler. The o-LDA first applies the batch Gibbs sampler to the full dataset and then samples new topics for each newly observed word using information observed so far. Canini et al. [13] extended o-LDA and proposed an incremental Gibbs sampler for LDA (here referred to as I-LDA). The I-LDA does not need a batch initialization phase like o-LDA. In o-LDA and I-LDA, the number of categories is fixed, while in our approach the number of categories is growing. Moreover, o-LDA and I-LDA are used to discover topics shared among all categories, while our approach is used to discover specific topics per category.

Currently, a popular approach in object recognition is deep learning. However, there are several limitations to use Deep Neural Networks (DNN) in open-ended domains. Deep networks are incremental by nature but not open-ended, since the inclusion of novel categories enforces a restructuring in the topology of the network. Moreover, DNN usually needs a lot of training data and long training times to obtain an acceptable accuracy. Schwarz et.al [14] used DNN for 3D object category learning. They clearly showed that the performance of DNN degrades when the size of dataset is reduced.

## 3  System overview

The main motivation of this work is to achieve a multi-layered object representation that builds an increasingly complex object representation (see Fig. 1). Particularly, a statistical model is used to get structural semantic features from low-level feature co-occurrences. The basic idea is that each object view is described as a random mixture over a set of latent topics, and each topic is defined as a discrete distribution over visual words (i.e. local shape features). It must be pointed out that we are using shape features rather than semantic properties to encode the statistical structure of object categories [15]. It is easier to explain the details using an example. We start by selecting a category label, for example *Mug*. To represent a new instance of Mug, a distribution over Mug topics is drawn that will specify which intermediate topics should be selected for generating each visual words of the object. According to this distribution, a particular topic is selected out of the mixture of possible topics of the Mug category for generating each visual word in the object. For instance, a *Mug* usually has a handle, and a "handle" topic refers to some visual words that occur frequently together in handles. The process of drawing both the topic and visual word is repeated several times to choose a set of visual words that would construct a *Mug*. We use statistical inference techniques for inverting this process to automatically find out a set of topics for each category from a collection of instances. In other words, we try to learn a model for each category (a set of latent variables) that explains how each object obtains its visual words. In our approach, the characteristics of surfaces belonging to objects are described by local shape features called spin-images [3].

## 4  Dictionary construction

Comparing 3D objects based on their local features is computationally expensive. The topic modelling approach directly addresses this concern. It requires a dictionary with $V$ visual words. Usually, the dictionary is created via off-line clustering of training data, while in open-ended learning, there is no training data available at the beginning of the learning process. To cope with this limitation, we propose that the robot freely explores several scenes and collects several object experiences.

In general, object exploration is a challenging task because of ill-definition of the objects [16]. Since a system of boolean equations can represent any expression or any algorithm, it is particularly well suited for encoding the world and object candidates. Similar to Collet's work [16], we have used boolean algebra based on the three logical operators, namely *AND* $\wedge$, *OR* $\vee$ and *NOT* $\neg$. A set of constraints, $C$, is then defined. Each constraint has been implemented as a function that returns either true or false (see Table 1).

Table 1: List of used constraints with a short description for each one.

| Constraints | Description |
| --- | --- |
| $C_{\text{table}}$: *"is this candidate on a table?"* | The interest object candidate is placed on top of a table. |
| $C_{\text{track}}$: *"is this candidate being tracked?"* | This constraint is used to infer that the segmented object is already being tracked or not. |
| $C_{\text{size}}$: *"is this candidate manipulatable?"* | Reject large object candidate. |
| $C_{\text{instructor}}$: *"is this candidate part of the instructor's body?"* | Reject candidates that are belong to the user's body. |
| $C_{\text{robot}}$: *"is this candidate part of the robot's body?"* | Reject candidates that are belong to the robot's body. |
| $C_{\text{edge}}$: *"is this candidate near to the edge of the table?"* | Reject candidates that are near to the edge of the table. |
| $C_{\text{key\_view}}$: *"is this candidate a key view?"* | Only key-views are stored into *Perceptual Memory*. |

Note that, storing all object views while the object is static would lead to unnecessary accumulation of highly redundant data. Therefore, $C_{\text{key\_view}}$ is used to optimize memory usage and computation while keeping potentially relevant and distinctive information. An object view is selected as a key view whenever the tracking of an object is initialized ($C_{\text{track}}$), or when it becomes static again after being moved. In case the hands are detected near the object, storing key views are postponed until the hands are withdrawn [17]. Using these constraints, boolean expressions, $\psi$, are built to encode object candidates for the *Object Exploration* and *Object Recognition* purposes (see equations 1 and 2).

$$\psi_{\text{exploration}} = C_{\text{table}} \wedge C_{\text{track}} \wedge \ C_{\text{key\_view}} \wedge \neg(C_{\text{instructor}} \vee C_{\text{robot}}), \tag{1}$$

$$\psi_{\text{recognition}} = C_{\text{table}} \wedge C_{\text{track}} \wedge \neg \ (C_{\text{instructor}} \vee C_{\text{robot}} \vee C_{\text{edge}}), \tag{2}$$

The basic perception infrastructure, which is strongly based on the Point Cloud Library (PCL), has been described in detail in previous publications [18][19]. A table is detected by finding the dominant plane in the point cloud. This is done using the RANSAC algorithm. The extraction of polygonal prisms mechanism is used for collecting the points which lie directly above the table. Afterwards, an Euclidean Cluster Extraction algorithm is used to segment each scene into individual clusters. Every cluster that satisfies the exploration expression is selected. The output of this object exploration is a pool of object candidates. Subsequently, to construct a pool of features, spin-images [3] are computed for the selected points extracted from the pool of object candidates. We computed around 32000 spin-images from the point cloud of the 194 objects views. Finally, the dictionary is constructed by clustering the features using the k-means algorithm. The centers of the $V$ extracted clusters are used as visual words, $\mathbf{w}_t$ ($1 \leq t \leq V$). A video of the robot exploring an environment[1] is available at: https://youtu.be/MwX3J6aoAX0.

## 5 Object representation

A hierarchical system is presented which follows the organization of the visual cortex and builds an increasingly complex object representation. Plasticity and learning can occurr at all layers and certainly at the top-most layers of the hierarchy. In this paper, object view representation in the *feature layer* involves two main phases: keypoint extraction and computation of spin images for the keypoints. For keypoint extraction, a voxelized grid approach is used to obtain a smaller set of points by taking only the nearest neighbor point for each voxel center. Afterwards, the spin-image descriptor is used to encode the surrounding shape in each keypoint using the original point cloud (i.e. *feature layer*). Subsequently, the spin images go "up" to the *BoW layer* where each spin image is assigned to a visual word by searching for the nearest neighbor in the dictionary. Afterwards, each object is represented as a set of visual words. The obtained representation is then presented as input to the *topic layer*. The LDA model consists of three levels' parameters including category-level parameters (i.e. $\alpha$), which are sampled once in the process of generating a category of objects; object-level variables (i.e. $\theta_d$), which are sampled once per object, and word-level variables (i.e. $\mathbf{z}_{d,n}$ and $w_{d,n}$), which are sampled every time a feature is extracted. The variables $\theta$, $\phi$ and $\mathbf{z}$ are latent variables that should be inferred. Assume everything is observed and a category label is selected for each object; i.e. each object belongs to one category. The joint distribution of all hidden and observed variables for a category is defined as follows:

$$p^{(c)}(\mathbf{w}, \mathbf{z}, \theta, \phi | \alpha, \beta) = \prod_{z=1}^{K} p^{(c)}(\phi_z | \beta) \prod_{d=1}^{|c|} p^{(c)}(\theta_d | \alpha) \prod_{n=1}^{N} p^{(c)}(\mathbf{z}_{d,n} | \theta_d) p^{(c)}(w_{d,n} | \mathbf{z}_{d,n}, \phi), \tag{3}$$

where $\alpha$ and $\beta$ are Dirichlet prior hyper-parameters that affect the sparsity of distributions, and $K$ is the number of topics, $|c|$ is the number of known objects in the category $c$ and $N$ is the number of words in the object $d$. Each $\theta_d$ represents an instance of category $c$ in topic-space as a Cartesian histogram (i.e. *topic layer*), $\mathbf{w}$ represents an object as a vector of visual words, $\mathbf{w} = \{w_1, w_2, ..., w_N\}$, where each entry represents one of the $V$ words of the dictionary (i.e. *BoW layer*). $\mathbf{z}$ is a vector of topics and $z_i = 1$ means $w_i$ was generated form $i^{th}$ topic. It should be noticed that there is a topic for each word and $\phi$ is a $K \times V$ matrix, which represents word-probability matrix for each topic, where $V$ is the size of dictionary and $\phi_{i,j} = p^{(c)}(w_i|z_j)$; thus, the posterior distributions of the latent variables given the observed data is computed as follows:

$$p^{(c)}(\mathbf{z}, \theta, \phi|\mathbf{w}, \alpha, \beta) = \frac{p^{(c)}(\mathbf{w}, \mathbf{z}, \theta, \phi|\alpha, \beta)}{p^{(c)}(\mathbf{w}|\alpha, \beta)}, \tag{4}$$

Unfortunately, the denominator of the equation 4 is *intractable* and can not be computed exactly. A collapsed Gibbs sampler is used to solve the inference problem. Since $\theta$ and $\phi$ can be derived from $z_i$, they are integrated out from the sampling procedure. In this work, for each category an incremental LDA model is created. Whenever a new training instance is presented, the collapsed Gibbs sampling is employed to update the parameters of the model. The collapsed Gibbs sampler is used to estimate the probability of topic $z_i$ being assigned to a word $w_i$, given all other topics assigned to all other words:

$$p^{(c)}(z_i = k|\mathbf{z}_{\neg i}, \mathbf{w}) \propto p^{(c)}(z_i = k|\mathbf{z}_{\neg i}) \times p^{(c)}(w_i|\mathbf{z}_{\neg i}, \mathbf{w}_{\neg i})$$

$$\propto \frac{n_{d,k,\neg i} + \alpha}{[\sum_{k=1}^{K} n_{d,k} + \alpha] - 1} \times \frac{n_{w,k,\neg i}^{(c)} + \beta}{\sum_{w=1}^{V} n_{w,k}^{(c)} + \beta}, \tag{5}$$

where $\mathbf{z}_{\neg i}$ means all hidden variables expect $z_i$ and $\mathbf{z} = \{z_i, \mathbf{z}_{\neg i}\}$. $n_{d,k}$ is the number of times topic $k$ is assigned to some visual word in object $d$ and $n_{w,k}^{(c)}$ shows the number of times visual word $w$ assigned to topic $k$. In addition, the denominator of the $p^{(c)}(z_i = k|\mathbf{z}_{\neg i})$ is omitted because it does not depend on $z_i$. The multinomial parameter sets $\theta^{(c)}$ and $\phi^{(c)}$ can be estimated using the following equations:

$$\theta_{k,d}^{(c)} = \frac{n_{d,k} + \alpha}{n_d + K\alpha}, \quad \text{and} \quad \phi_{w,k}^{(c)} = \frac{n_{w,k}^{(c)} + \beta}{n_k^{(c)} + V\beta}. \tag{6}$$

where $n_k^{(c)}$ is the number of times a word assigned to topic $k$ in category $c$ and $n_d$ is the number of words exist in the object $d$. Since in this approach, what happens next depends only on the current state of the system and not on the sequence of previous states, whenever a new object view, $\theta_d^{(c)}$, is added to the category $c$, $n_k^{(c)}$ and $n_{w,k}^{(c)}$ are updated incrementally.

## 6  Object category learning and recognition

Whenever a new object view is added to a category [17], the object conceptualizer retrieves the current model of the category as well as representation of the new object view, and creates a new, or updates the existing category. To exemplify the strength of object representation, an instance-based learning approach is used in the current system, i.e. object categories are represented by sets of known instances. The instance-based approach is used because it is a baseline method for category representation. However, more advanced approaches like Bayesian approach can be easily adapted. An advantage of the instance based approach is to facilitate incremental learning in an open-ended fashion. Similarly, a baseline recognition mechanism in the form of a nearest neighbour classifier with a simple thresholding approach are used to recognize a given object view.

The query object view, $\mathbf{O}_q$, is first represented using the topic distribution of each category, $\theta_q^{(c)}$. Afterwards, to assess the dissimilarity between the query object and stored instances of category $c$, $\theta_p$, the symmetric Kullback Leibler divergence, i.e. $\mathrm{D_{KL}}(\theta_q^{(c)}, \theta_p)$, is used to measure the difference between two distributions. Subsequently, the minimum distance between the query object and all instances of the category $c$, is considered as the Object-Category Distance, $\mathrm{OCD}(.)$:

$$\mathrm{OCD}(\theta_q^{(c)}, c) = \min_{\theta_p \in c} \mathrm{D_{KL}}(\theta_q^{(c)}, \theta_p), \quad c \in \{1, \ldots, C\}. \tag{7}$$

Consequently, the query object is classified based on the minimum OCD(.). If, for all categories, the OCD(.) is larger than a given Classification Threshold (e.g. CT= 0.75), then the object is classified as *unknown*; otherwise, it is classified as the category that has the highest similarity.

## 7 Experimental results

The proposed approach was evaluated using a standard cross-validation protocol as well as an open-ended protocol. We also report on a demonstration of the system.

### 7.1 Off-line evaluation

An object dataset has been used [18], which contains 339 views of 10 categories of objects. The system has five different parameters that must be well selected to provide a good balance between recognition performance and memory usage. To examine the performance of different configurations of the proposed approach, 10-fold cross-validation has been used. A total of 180 experiments were performed for different values of five parameters of the system, namely the voxel size (VS), which determines the number of keypoints extracted from each object view, the image width (IW) and support length (SL) of spin images, the dictionary size (DS) and the number of topics (NT). Results are presented in Table 2. The parameters that obtained the best average accuracy was selected as the default configuration: VS=0.03, IW=4 and SL=0.05, DS=90 and NT=30. In all experiments, the number of iterations for Gibbs sampling was 30 and $\alpha$ and $\beta$ parameters were set to 1 and 0.1 respectively. The accuracy of the proposed system with the default configuration was $0.87$. Therefore, this configuration displays a good balance between recognition performance and memory usage. The remaining results were obtained using this configuration.

The accuracy of the system in each layer has been calculated individually. For comparison, the accuracy of a topic layer with topics shared among all categories is also computed. Results are presented in Table 3. One important observation is that the overall performance of the recognition system based on topic modelling is promising and the proposed

Table 3: Object recognition performance

| Representation | Accuracy |
|---|---|
| Feature Layer | 0.12 |
| BoW Layer | 0.79 |
| Topic Layer (shared topics) | 0.79 |
| Topic Layer (our approach) | 0.87 |

representation is capable of providing distinctive representation for the given object. Moreover, it was observed that the discriminative power of the proposed representation was better than the other layers. In addition, independent topics for each category provides better representation than shared topics for all categories. Furthermore, it has been observed that the discriminative power of shared topics depends on the order of introduction of categories.

The accuracy of object recognition based on pure shape features (i.e. feature layer) is very low. The BoW representation obtains an acceptable performance. The *topic layer* provides a good balance between memory usage and descriptiveness with 30 floats (i.e. NT=30). The length of the BoW layer is around three times larger than the representation of the *topic layer*. The *feature layer* is the less compact representation. These results show the hierarchical object representation builds an increasingly complex representation.

### 7.2 Open-ended evaluation

The off-line evaluation methodologies (e.g k-fold cross validation, etc.) are not well suited to evaluate open-ended learning systems, because they do not abide to the simultaneous nature of learning and recognition. Those methodologies imply that the set of categories must be predefined. An evaluation protocol for open-ended learning systems was proposed in [20]. The idea is to emulate the interactions of a recognition system with the surrounding environment over long periods of time. A simulated teacher was developed to follow the evaluation protocol and autonomously interact with the recognition system using three basic actions including: ***teach***, for teaching a new object category; ***ask***, to ask the system what is the category of an object view; and ***correct***, for providing

Table 2: Object recognition performance for different parameters

| Parameters | VS(m) | | IW (bins) | | SL(m) | | | DS (visual words) | | | | | NT | | |
|---|---|---|---|---|---|---|---|---|---|---|---|---|---|---|---|
| Values | 0.03 | 0.04 | 4 | 8 | 0.04 | 0.05 | 0.06 | 50 | 60 | 70 | 80 | 90 | 30 | 40 | 50 |
| Avg. Accuracy(%) | 85 | 81 | 83 | 83 | 82 | 83 | 83 | 82 | 82 | 83 | 84 | 84 | 84 | 83 | 82 |

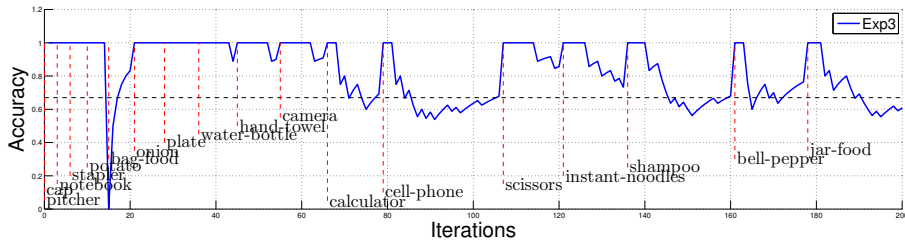

Figure 2: Evolution of accuracy vs. number of question/correction iterations in the first 200 iterations of the third experiment. Vertical red lines and labels indicate when and which categories are introduced to the system.

corrective feedback, i.e. the ground truth label of a misclassified object view. The idea is that, for each newly taught category, the simulated teacher repeatedly picks unseen object views of the currently known categories from a dataset and presents them to the system. It progressively estimates the recognition accuracy of the system and, in case this accuracy exceeds a given threshold (marked by the horizontal line in Fig.2), introduces an additional object category (marked by the vertical lines and labels in Fig.2). This way, the system is trained, and at the same time the accuracy of the system is continuously estimated. The simulated teacher must be connected to an object dataset. In this work, the simulated teacher was connected to the largest available dataset namely RGB-D Object Dataset consisting of 250,000 views of 300 common household objects, organized into 51 categories [21].

Since the performance of an open-ended learning system is not limited to the object recognition accuracy, when an experiment is carried out, learning performance is evaluated using three distinct measures, including: (*i*) the number of learned categories at the end of an experiment

Table 4: Summary of experiments.

| EXP# | #QCI | #TLC | #AIC | GCA (%) | APA (%) |
|------|------|------|------|---------|---------|
| 1 | 1740 | 39 | 18.38 | 65 | 71 |
| 2 | 803 | 30 | 11.07 | 69 | 79 |
| 3 | 1099 | 35 | 13.20 | 67 | 77 |
| 4 | 1518 | 38 | 16.29 | 66 | 73 |
| 5 | 1579 | 42 | 15.12 | 67 | 72 |

(TLC), an indicator of *How much does it learn?*; (*ii*) The number of question / correction iterations (QCI) required to learn those categories and the average number of stored instances per category (AIC), indicators of *How fast does it learn?* (see Fig.3 (right)); (*iii*) Global classification accuracy (GCA), an accuracy computed using all predictions in a complete experiment, and the average protocol accuracy (APA), indicators of *How well does it learn?* (see Fig.3 (left)). Since the order of the categories introduced may have an affect on the performance of the system, five experiments were carried out in which categories were introduced in random sequences. Results are reported in Table 4. Figure 2 shows the performance of the system in the initial 200 iterations of the third experiment. By comparing all experiments, it is visible that in the fifth experiment, the system learned more categories than other experiments. Figure 3 (*left*) shows the global classification accuracy obtained by the proposed approach as a function of the number of learned categories. In experiments 1, 4, 5, the accuracy first decreases, and then starts slightly going up again as more categories are introduced. This is expected since the number of categories known by the system makes the classification task more difficult. However, as the number of learned categories increases, also the number of instances per category increases, which augments the category models (topics) and therefore improves performance of the system. Fig.3 (*right*) gives a measure of how fast the learning occurred in each of the experiments and shows the number of question/correction iterations required to learn a certain number of categories. Our approach learned faster than that of Schwarz et. al [14] approach, i.e. our approach requires much less examples than Schwarz's work. Furthermore, we achieved accuracy around 75% while storing less than 20 instances per category (see Table 4), while Schwarz et.al [14] stored more than 1000 training instances per category (see Fig.8 in [14]). In addition, they clearly showed the performance of DNN degrades when the size of dataset is reduced.

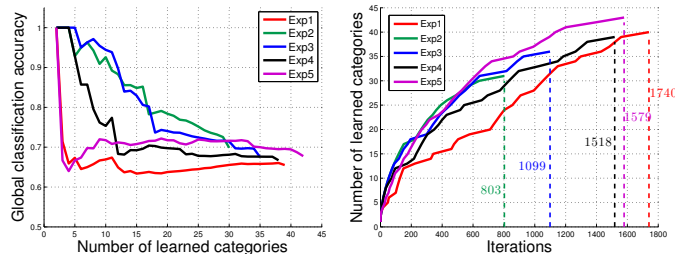

Figure 3: System performance during simulated user experiments.

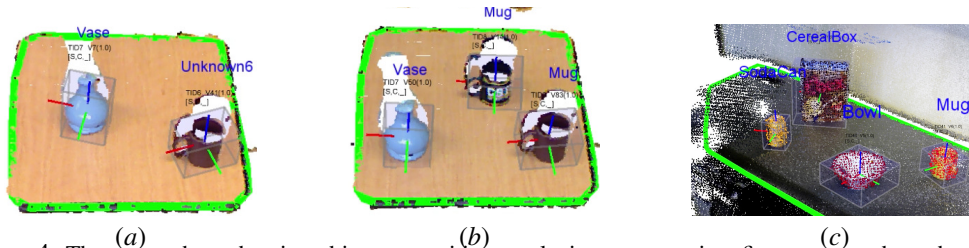

(a)                                    (b)                                    (c)

Figure 4: Three snapshots showing object recognition results in two scenarios: first two snapshots show the proposed system supports (*a*) classical learning from a batch of train labelled data and (*b*) open-ended learning from on-line experiences. Snapshot (*c*) shows object recognition results on a scene of Washington scene dataset.

### 7.3 System demonstration

To show the strength of object representation, a real demonstration was performed, in which the proposed approach has been integrated in the object perception system presented in [18]. In this demonstration a table is in front of a robot and two users interact with the system. Initially, the system only had prior knowledge about the *Vase* and *Dish* categories, learned from batch data (i.e. set of observations with ground truth labels), and there is no information about other categories (i.e. *Mug*, *Bottle*, *Spoon*). Throughout this session, the system must be able to recognize instances of learned categories and incrementally learn new object categories. Figure4 illustrates the behaviour of the system:

(a) The instructor puts object TID6 (a *Mug*) on the table. It is classified as *Unknown* because mugs are not known to the system; Instructor labels TID6 as a *Mug*. The system conceptualizes *Mug* and TID6 is correctly recognized. The instructor places a *Vase* on the table. The system has learned *Vase* category from batch data, therefore, the *Vase* is properly recognized (Fig.4 (*a*)).

(b) Later, another *Mug* is placed on the table. This particular *Mug* had not been previously seen, but the system can recognize it, because the Mug category was previously taught (Fig.4 (*b*)).

This demonstration shows that the system is capable of using prior knowledge to recognize new objects in the scene and learn about new object categories in an open-ended fashion. A video of this demonstration is available at: https://youtu.be/J0QOc_Ifde4.

Another demonstration has been performed using Washington RGB-D Scenes Dataset v2. This dataset consists of 14 scenes containing a subset of the objects in the RGB-D Object Dataset, including bowls, caps, mugs, and soda cans and cereal boxes. Initially, the system had no prior knowledge. The four first objects are introduced to the system using the first scene and the system conceptualizes those categories. The system is then tested using the second scene of the dataset and it can recognize all objects except cereal boxes, because this category was not previously taught. The instructor provided corrective feedback and the system conceptualized the cereal boxes category. Afterwards, all objects are classified correctly in all 12 remaining scenes (Fig.4 (c)). This evaluation illustrates the process of acquiring categories in an open-ended fashion. A video of this demonstration is online at: https://youtu.be/pe29DYNolBE.

## 8 Conclusion

This paper presented a multi-layered object representation to enhance a concurrent 3D object category learning and recognition. In this work, for optimizing the recognition process and memory usage, each object view was hierarchically described as a random mixture over a set of latent topics, and each topic was defined as a discrete distribution over visual words. This paper focused in detail on unsupervised object exploration to construct a dictionary and concentrated on supervised open-ended object category learning using an extension of topic modelling. We transform objects from bag-of-words space into a local semantic space and used distribution over distribution representation for providing powerful representation and deal with the semantic gap between low-level features and high-level concepts. Results showed that the proposed system supports classical learning from a batch of train labelled data and open-ended learning from actual experiences of a robot.

### Acknowledgements

This work was funded by National Funds through FCT project PEst-OE/EEI/UI0127/2016 and FCT scholarship SFRH/BD/94183/2013.

## Footnotes

[1]The ROS bag file used in this video was created by the Knowledge-Based Systems Group, Institute of Computer Science, University of Osnabrueck.

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
