[Reviews · NeurIPS 2016]

Reviewer 1

Summary

The paper describes a novel hierarchical learning architecture for recognising object categories from images that might be used e.g. by a robot. The system combines unsupervised and supervised learning, allowing it to flexibly learn new object categories as they are encountered. Demonstrations show that the system performs well in a real-world setting.

Qualitative Assessment

I have read the rebuttal, and it answered my specific questions. There's no change to my review. -- I'm unfamiliar with the application area, and while the problem tackled here is to my mind very relevant and challenging, I can't really say how large the new contributions here are. The demonstration videos linked in the paper were very useful in showing the capabilities of the system, but I don't know how that compares to state of the art. The experimental results reported in the paper are promising, but there are only a couple of lines comparing them to previous results. To summarise, the contribution seems to me novel and significant, but I have very low confidence here. The models and learing algorithms used as components of the system seem fairly standard, but finding and choosing the right tools and putting them together, taking into account both the specific application area (3D object recognition) and more general issues in open-ended hierarchical learning, is certainly a contribution by itself. lines 216-217: Here you say a category is represented by a set of instances, but Equation (3) looks like you represent a category as a single probability distribution. Perhaps you could clarify this. Figure 2 could use a bit more explanation. In particular, I don't understand the meaning of the specific placements of the category labels.

Confidence in this Review

1-Less confident (might not have understood significant parts)


Reviewer 2

Summary

The authors present an open-ended 3D object recognition system that learns the object categories and semantic features simultaneously. They use a topic model, Latent Dirichlet Allocation, to learn semantic features from low-level features for each object category. The authors conduct an off-line experiment and an open-ended evaluation. They also show a system demonstration.

Qualitative Assessment

* Line 39-41, the reasons for using topic modelling seems insufficient. Although LDA seems to provide short descriptions, but we still need big memory to train LDA model for each object view (from BOW layer (this needs big memory.) to topic layer). Maybe the authors need to clearly and quantitatively show that LDA is the best choice for optimizing memory. * Line 159, "RANSAC algorithm", please add the reference citation. * Line 164, "PCL function", please add the full name and add the reference citation. * Line 225-226, this object-category distance determination is kind of kNN method (k=1). Why do you choose k=1 and what is the effect of different k values? * Line 228, how to select the threshold? Do you think that a fixed threshold is always ok? what is its influence? Errors might incrementally accumulate. This relates to the order of the categories introduced. The results in Figure 3 describe this situation. I think this is a big issue in this approach. * Line 243, 0.87 percent (typo?) * Why LDA is important in this paper? * On line 307-310, in Comparison with deep learning, the proposed approach needs many question/correction iterations to learn categories. It is unfair to just compare these two methods with the number of training instances. * The proposed problem seems to that the robots can only sequentially recognize 3D objects. For an object of not existing category, if we often put it as a new category, what is the influence on the approach. In real world, it is not possible to expect the robots always recognize objects in order, the authors need to explain this situation.

Confidence in this Review

2-Confident (read it all; understood it all reasonably well)


Reviewer 3

Summary

In the “Introduction” section, the authors point out the neurophysiological evidences that a human brain has a hierarchical structure for the object recognition, and that the learning and recognition of objects occurs concurrently in a human brain. Then, they briefly explained how they built their 3D object recognition model that concurrently learns and recognizes the objects that works in the similar way to the human brain. The authors talk about conventional approaches for the object recognition problem in the “Related work” section. They mention about the probabilistic latent semantic indexing (pLSI), latent Dirichlet allocation (LDA) and its variations, etc. And they point out that none of these conventional models can learn and recognize objects in an open-ended manner. It was explained in the section, that an open-ended system is a system that can concurrently learn and recognize objects. They also mentioned that the number of classes for an open-ended system is not predefined and can grow continuously. For section 3 to 6, the authors explained their proposed model. The model is designed to use the LDA in recognizing as well as in learning both categories and their features in an open-ended manner. To do this, they used 5 layers to build their system: feature layer, Bag-of-Words (BoW) layer, topic layer, view layer, category layer. The feature layer computes key points from spin-images. Then the BoW layer assigns a visual word for each spin-images. The topic layer uses LDA to discover abstract topics. Then the view layer stores the random mixture over latent topics of categories. Finally, the categorical output is made in the category layer. The authors show their 3 experimental results at section 7. Their model was tested with a dataset using 10-fold cross validation method. The results showed that their topic layer, which develops independent topics for each category, showed better performance than the conventional approach (supervised LDA, class LDA) of sharing topics. And also showed that the test accuracy obtained from BoW layer is higher than the feature layer. Second experiment was conducted in an open-ended manner and showed a good performance of the model in recognizing and learning object categories concurrently. Finally, in the last experiment, the authors demonstrated how the model can work for robots with the aid of human instructors.

Qualitative Assessment

For the first experiment, the authors should have compared their model with a variety of models. There are many deep learning models for 3D object recognition, yet, their performances were not compared with that of the authors’ model when their model was tested in an off-line condition. Therefore I evaluated the technical quality of this paper as “Poster level”. In the paper, the authors used abbreviations such as PCL, OCD without explaining what they stand for. Therefore I gave “Poster level” for the technical quality. The proposed model has novelty in that it recognizes and learns the category of 3D objects concurrently using latent Dirichlet allocation. By using the model, the tedious work of making dataset, and training a robot with it to adapt to its working environment may be alleviated for the researchers in the robotics field. Therefore, I believe that this study has made an advance in the field of vison processing for the robotics. Therefore I evaluated both novelty and potential usefulness as “Oral level”.

Confidence in this Review

1-Less confident (might not have understood significant parts)


Reviewer 4

Summary

The research paper discusses way to jointly learn about objects and their categories by using a modified version of LDA. The categories are learned in an unsupervised manner. The framework tries to solve the problem of joint learning of images and their associated language in open-ended manner i.e. the system starts with little prior knowledge.

Qualitative Assessment

1. A potential missing reference is "A Joint Model of Language and Perception for Grounded Attribute Learning" by Cynthia Matuszek. 2. There are few English grammatical mistakes such as paragraph 1 (Introduction) "order that generates recognition result at the end" can be worded more clearly. 3. The author can differentiate between online training and open-ended training where online training is fixed set of labels but the model trains while performing predictions. Open-ended training is where the model learns about unknown labels. A survey of online training methods might build the authors case better. 4. The author does not provide sufficient details about when and how minimum ground knowledge (i.e. if minimum set of labels for objects are provided as prior from which the system begins to learn). Page 2 Related Work: 1. "while we propose to learn specific topics per category". Does the author mean unique topics? Page 3 System Overview: The author can define what a visual word is. Page 4 Object representation The spin image is assigned to a visual word based on the nearest neighbor in the dictionary. This according to the paper may be empty at the beginning. The open-end system evaluation uses an external person to provide answers to queries (an active learning approach). How the answers are incorporated into the system is unclear (or can be written with more clarity). Page 5 Object representation "what happens next depends only on the current state of the system". The author can define the state and what is meant by sequence of previous states. Page 6 CT is undefined. Page 6 Experimental Results Table 3, it is unclear as to how the accuracy for each layer was measured (as the outputs for each layer differ?!). Also if each layer feeds into the next layer its unclear as the why the accuracy increases (since bad feature representations from one layer in pipeline will affect the next ?!). Figure 3 The accuracies change over each new category of object, does the same trend hold true for all layers (if other layer accuracies were measured like in Figure 3). A discussion on the final size of the dictionary, performance challenges especially when the number of latent topics becomes large in number. The system is also not hierarchical in sense that it does not seem to construct a hierarchy among the object categories. The system has a hierarchal way of building a representation of object categories.

Confidence in this Review

2-Confident (read it all; understood it all reasonably well)